

# Metallic and insulating stripes and their relation with superconductivity in the doped Hubbard model

Luca F. Tocchio[1*], Arianna Montorsi[1], Federico Becca[2]

**1** Institute for Condensed Matter Physics and Complex Systems,
DISAT, Politecnico di Torino, I-10129 Torino, Italy
**2** Dipartimento di Fisica, Università di Trieste, Strada Costiera 11, I-34151 Trieste, Italy

★ luca.tocchio@polito.it

## Abstract

The dualism between superconductivity and charge/spin modulations (the so-called stripes) dominates the phase diagram of many strongly-correlated systems. A prominent example is given by the Hubbard model, where these phases compete and possibly co-exist in a wide regime of electron dopings for both weak and strong couplings. Here, we investigate this antagonism within a variational approach that is based upon Jastrow-Slater wave functions, including backflow correlations, which can be treated within a quantum Monte Carlo procedure. We focus on clusters having a ladder geometry with $M$ legs (with $M$ ranging from 2 to 10) and a relatively large number of rungs, thus allowing us a detailed analysis in terms of the stripe length. We find that stripe order with periodicity $\lambda = 8$ in the charge and $2\lambda = 16$ in the spin can be stabilized at doping $\delta = 1/8$. Here, there are no sizable superconducting correlations and the ground state has an insulating character. A similar situation, with $\lambda = 6$, appears at $\delta = 1/6$. Instead, for smaller values of dopings, stripes can be still stabilized, but they are weakly metallic at $\delta = 1/12$ and metallic with strong superconducting correlations at $\delta = 1/10$, as well as for intermediate (incommensurate) dopings. Remarkably, we observe that spin modulation plays a major role in stripe formation, since it is crucial to obtain a stable striped state upon optimization. The relevance of our calculations for previous density-matrix renormalization group results and for the two-dimensional case is also discussed.



# 1 Introduction

Despite a long time since the first experimental evidence of superconductivity in copper-oxide materials [1], the phase diagram of high-temperature cuprate superconductors still contains many unsolved questions. One of them is the possible coexistence or competition of superconductivity with inhomogeneous orders in the charge and spin sectors [2–4]. Starting from the neutron scattering observation of half-filled stripes in $La_{1.48}Nd_{0.4}Sr_{0.12}CuO_4$ around doping 1/8 [5], different inhomogeneous orders have been proposed with a variety of experimental probes, ranging from scanning tunneling microscopy to x-ray scattering [6–9].

From a theoretical point of view, striped states were first considered in the strong-coupling limit, i.e., within the so-called $t-J$ model. Here, contradictory results have been obtained with different numerical methods. In particular, half-filled stripes are found to be stable by the density-matrix renormalization group (DMRG) [10], the infinite projected entangled-pair states (iPEPS) [11], and variational Monte Carlo [12] approaches. However, improved variational wave functions, with the inclusion of hole-hole repulsion [13], as well as recent calculations based upon the renormalized mean-field theory [14], show that striped and homogeneous states are almost degenerate. The absence of stripes has been reported by further improvements of variational Monte Carlo, including the application of a few Lanczos steps and an imaginary-time projection performed within a fixed-node approximation [15]. Even though the $t-J$ model has been widely used in the past to describe the low-energy physics of cuprate materials, neglecting local charge fluctuations is not completely justified; in addition, antiferromagnetic correlations are usually overestimated, because of the presence of a direct super-exchange term. These two ingredients may have an important impact on the possible stabilization of charge and/or spin modulations at finite dopings. In this respect, a better description can be obtained by considering the Hubbard model:

$$\mathcal{H} = -t \sum_{\langle R,R'\rangle,\sigma} c^{\dagger}_{R,\sigma} c_{R',\sigma} + \text{H.c.} + U \sum_R n_{R,\uparrow} n_{R,\downarrow}, \tag{1}$$

where $c^{\dagger}_{R,\sigma}$ ($c_{R,\sigma}$) creates (destroys) an electron with spin $\sigma$ on site $R$ and $n_{R,\sigma} = c^{\dagger}_{R,\sigma} c_{R,\sigma}$ is the electronic density per spin $\sigma$ on site $R$. In the following, we indicate the coordinates of the sites with $\mathbf{R} = (x, y)$. The nearest-neighbor hopping integral is denoted as $t$, while $U$ is the on-site Coulomb interaction. The electronic doping is given by $\delta = 1 - N_e/L$, where $N_e$ is the number of electrons (with vanishing total magnetization) and $L$ is the total number of sites. For ladder systems, which will be considered in the paper, we define $M$ as the number of legs and $L_x$ as the number of rungs, the total number of sites being given by $L = M \times L_x$.

For positive $U$ and an even number of legs $M$, the formation of short-ranged antiferromagnetic electron pairs is responsible for the opening of a spin gap in a finite doping range $0 < \delta \leq \delta_c$. This is signaled by the hidden ordering of a spin-parity operator [16, 17]. From one side, this hidden ordering is accompanied by large superconducting correlations for $M \geq 4$ [18], suggesting a direct connection with the $d$-wave superconductivity observed in the uniform state of the two-dimensional Hubbard model [19–26]. From the other side, stripes may be stabilized, with a possible suppression of the superconducting order. Indeed, early DMRG studies suggested the formation of stripes on the 6-leg Hubbard model [27, 28]; these results have been corroborated by using constrained path auxiliary-field Monte Carlo [29] and density-matrix embedding theory [30]. More recently, a combination of advanced numerical methods [31] focused on doping 1/8, where maximal inhomogeneity is observed in many high-temperature superconductors. This work highlighted the fact that the ground state is a bond-centered linear stripe, with periodicity $\lambda = 8$ in the charge (which defines the wavelength of the stripe) and periodicity $2\lambda = 16$ in the spin, due to the so-called $\pi$-phase shift; in this case, the stripe is *filled* and commensurate with the doping, since it accommodates

two holes in a one-dimensional cell of length 16. This striped state is reported to have an energy lower of about $0.01t$ with respect to the uniform state. Other states, such as diagonal stripes or checkerboard order have been tested, but they were not found to be competitive with the linear stripe. Even if different stripe wavelengths are close in energy, the most favorable one is the filled one with wavelength $\lambda = 8$, in agreement with older Hartree-Fock calculations [32–35], but unlike the stripes in real materials, that are half-filled with wavelength $\lambda = 4$. The presence of nearest-neighbor $d-$wave coupling coexisting with the stripe has been also reported in this study [31]. A striped nature of the ground state, also away from the 1/8 doping, has been suggested by both cellular dynamical mean-field theory [36] and variational Monte Carlo calculations [37,38]. Both methods report the presence of stripes in the doping range $0.07 \lesssim \delta \lesssim 0.17$, with the ground state being uniform for higher dopings; the stripe wavelength is shown to decrease when the doping increases, with the optimal stripe wavelength at doping 1/8 being slightly different from what reported in Ref. [31]. Interestingly, a finite superconducting order parameter at large distances has been reported to coexist with stripes in Ref. [38]. Concerning the wavelength of the stripe, it has been suggested by a finite-temperature determinant quantum Monte Carlo study that the presence of a finite next-nearest-neighbor hopping $t'$ reduces the stripe wavelength, from $\lambda = 8$ at $t' = 0$ to $\lambda = 5$ at $t'/t = -0.25$ [39].

In this paper, we employ the variational Monte Carlo method with backflow correlations to study the presence of striped ground states in the doped Hubbard model. First, we consider doping $\delta = 1/8$ on ladders with $M = 2$, 6, and 10, in order to extrapolate to the two-dimensional limit. While no evidence of static stripes is seen on the 2-leg ladder, a filled stripe of wavelength $\lambda = 8$ is stabilized on a larger number of legs, as well in the two-dimensional extrapolation. This state is compatible with an insulator, with charge and spin order and no superconductivity. On a 6-leg ladder, we have investigated also the dopings away from $\delta = 1/8$. In particular, at doping $\delta = 1/6$, a striped state of wavelength $\lambda = 6$, compatible with an insulating state, occurs, while, at doping $\delta = 1/12$, the best striped state is not filled, having wavelength $\lambda = 8$. However, this state is only weakly metallic, since, on a 6-leg ladder, it is commensurate with the doping, having an even number of holes per spin modulation. On the contrary, away from these commensurate dopings, stripes can be metallic and coexisting with superconductivity, even if it is suppressed with respect to the homogeneous case. In this respect, we show that in the whole range between doping $\delta = 1/12$ and $\delta = 1/8$, the best ground-state approximation is a striped state with wavelength $\lambda = 8$, but with a metallic behavior and superconducting correlations with a power-law decay. Furthermore, we report that spin modulation plays a major role in stripe formation, since it is crucial to obtain a stable striped state upon optimization. The role of spin modulation can be also observed in the formation of a peak in spin-spin correlations, that is much stronger than the one in density-density correlations. In summary, our results show that striped states are a common feature of the doped repulsive Hubbard model. They coexist with superconductivity away from selected dopings where they are commensurate, their formation being favored by the presence of magnetic order with $\pi$ shift.

## 2   Variational Monte Carlo method

Our numerical results are obtained by means of the variational Monte Carlo method, which is based on the definition of suitable wave functions to approximate the ground-state properties beyond perturbative approaches [40]. In particular, we consider the so-called Jastrow-Slater wave functions that include long-range electron-electron correlations, via the Jastrow factor [41, 42], on top of uncorrelated states, extending the original formulation proposed

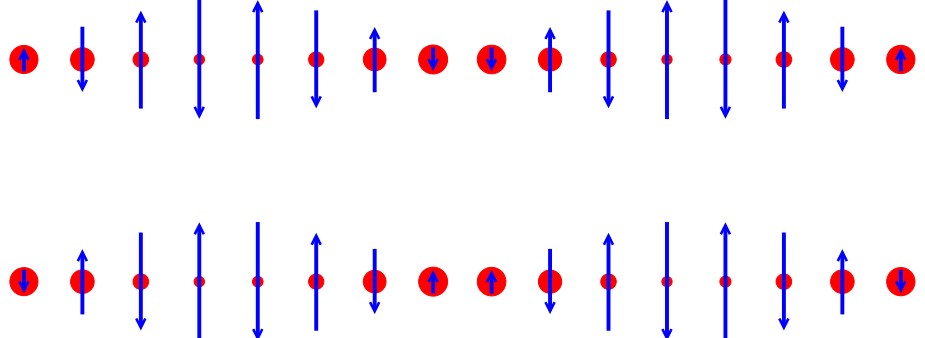

Figure 1: Unit cell for a stripe of wavelength $\lambda = 8$. The size of the red circles is proportional to the hole density per site, while the length of the arrows is proportional to the magnetization per site. Data are taken from the optimal state at $U/t = 8$ and $\delta = 1/8$.

by Gutzwiller [43, 44]. In addition, the so-called backflow correlations will be applied to the Slater determinant [45, 46], in order to sizably improve the quality of the variational state. Our variational wave functions are described by:

$$|\Psi\rangle = \mathcal{J}|\Phi_0\rangle, \tag{2}$$

where $\mathcal{J}$ is the Jastrow factor and $|\Phi_0\rangle$ is a state that, starting from an uncorrelated wave function obtained from an auxiliary Hamiltonian, redefines the orbitals on the basis of the many-body electronic configuration, incorporating virtual hopping processes, as discussed in Refs. [45, 46]. Unless otherwise specified, results are obtained considering backflow corrections.

We consider two different kinds of wave functions, the first one that is appropriate to study homogeneous superconducting states (with possible additional Néel antiferromagnetism) and the second one that describes bond-centered striped states with charge and spin modulation. The latter one can be further supplemented with either uniform or modulated pairing.

Let us start with the homogeneous superconducting state that is the ground state of the following auxiliary Hamiltonian:

$$\mathcal{H}_{\mathrm{BCS}} = \sum_{k,\sigma} \epsilon_k c_{k,\sigma}^{\dagger} c_{k,\sigma} - \mu \sum_{k,\sigma} c_{k,\sigma}^{\dagger} c_{k,\sigma} + \sum_k \Delta_k c_{k,\uparrow}^{\dagger} c_{-k,\downarrow}^{\dagger} + \text{h.c.}, \tag{3}$$

which includes a free-band dispersion $\epsilon_k$, defined as in the Hubbard Hamiltonian of Eq. (1), a BCS coupling $\Delta_k$, with nearest-neighbor pairings $\Delta_x$ and $\Delta_y$ along the $x$ and the $y$ direction, respectively, and a chemical potential $\mu$. Néel antiferromagnetism can be further included by adding the following term to the Hamiltonian of Eq. (3):

$$\mathcal{H}_{\mathrm{AF}} = \Delta_{\mathrm{AF}} \sum_R (-1)^{x+y} \left( c_{R,\uparrow}^{\dagger} c_{R,\uparrow} - c_{R,\downarrow}^{\dagger} c_{R,\downarrow} \right). \tag{4}$$

All the parameters $\Delta_x$, $\Delta_y$, $\Delta_{\mathrm{AF}}$, and $\mu$ are optimized to minimize the variational energy (while $t = 1$, in the definition of $\epsilon_k$, sets the energy scale of the uncorrelated Hamiltonian).

Bond-centered striped states can be included in the variational wave function by means of charge and spin modulations along the $x$ direction. The auxiliary Hamiltonian reads as

$$\mathcal{H}_{\mathrm{MF}} = \mathcal{H}_{\mathrm{BCS}} + \mathcal{H}_{\mathrm{charge}} + \mathcal{H}_{\mathrm{spin}}. \tag{5}$$

Here, the term

$$\mathcal{H}_{\mathrm{charge}} = \Delta_c \sum_R \cos\left[Q\left(x - \frac{1}{2}\right)\right]\left(c_{R,\uparrow}^{\dagger} c_{R,\uparrow} + c_{R,\downarrow}^{\dagger} c_{R,\downarrow}\right) \tag{6}$$

describes a charge modulation in the $x$ direction with periodicity $\lambda = 2\pi/Q$; the term

$$\mathcal{H}_{\text{spin}} = \Delta_{\text{AF}} \sum_R (-1)^{x+y} \sin\left[\frac{Q}{2}\left(x - \frac{1}{2}\right)\right]\left(c^{\dagger}_{R,\uparrow}c_{R,\uparrow} - c^{\dagger}_{R,\downarrow}c_{R,\downarrow}\right) \tag{7}$$

describes an antiferromagnetic order that is the standard Néel one along the $y$ direction, while it is modulated along the $x$ direction with a periodicity that is doubled with respect to the charge one, i.e., $2\lambda = 4\pi/Q$; indeed, after one period in the charge modulation, spins are reversed, as illustrated in Fig. 1 for $\lambda = 8$. In the presence of stripes, $\Delta_c$ and $\Delta_{\text{AF}}$ are further variational parameters to be optimized.

The BCS term that is included in Eq. (5) can be either uniform with pairings $\Delta_x$ and $\Delta_y$ along the $x$ and the $y$ direction, respectively, or modulated with the following periodicity, that has been named "in phase" in Ref. [12]:

$$\Delta_{R,R+\hat{x}} = \Delta_x \left|\cos\left[\frac{Q}{2}x\right]\right| \qquad \Delta_{R,R+\hat{y}} = -\Delta_y \left|\cos\left[\frac{Q}{2}\left(x - \frac{1}{2}\right)\right]\right|, \tag{8}$$

where $\hat{x}$ and $\hat{y}$ are the versors along the $x$ and the $y$ directions, respectively, while $\Delta_x$ and $\Delta_y$ are variational parameters to be optimized. In Ref. [12], a different modulation in the pairing, named "antiphase" has been introduced, without the absolute values that are present in Eq. (8). However, in our simulation, we always found that the "antiphase" modulation does not lead to any energy gain with respect to the "in phase" case. We have also verified that the modulation proposed in Ref. [13], where hole density is maximum at the sites with the smallest pairing amplitude and the smallest magnetization, does not lead to an energy gain with respect to the modulation described in Eq. (8).

The Jastrow factor $J$ is defined as:

$$\mathcal{J} = \exp\left(-\frac{1}{2}\sum_{R,R'} v_{R,R'} n_R n_{R'}\right), \tag{9}$$

where $n_R = \sum_{\sigma} n_{R,\sigma}$ is the electron density on site $R$ and $v_{R,R'}$ (that include also the local Gutzwiller term for $\mathbf{R} = \mathbf{R}'$) are pseudopotentials that are optimized for every independent distance $|\mathbf{R} - \mathbf{R}'|$.

The presence of charge disproportionations can be analyzed by means of the static structure factor $N(\mathbf{q})$ defined as:

$$N(\mathbf{q}) = \frac{1}{L}\sum_{R,R'}\langle n_R n_{R'}\rangle e^{i\mathbf{q}\cdot(\mathbf{R}-\mathbf{R}')}, \tag{10}$$

where $\langle \ldots \rangle$ indicates the expectation value over the variational wave function. In particular, the presence of a peak at a given $\mathbf{q}$ vector denotes the presence of charge order in the system. Moreover, the static structure factor allows us to assess the metallic or insulating nature of the ground state. Indeed, charge excitations are gapless when $N(\mathbf{q}) \propto |\mathbf{q}|$ for $|\mathbf{q}| \to 0$, while a charge gap is present whenever $N(\mathbf{q}) \propto |\mathbf{q}|^2$ for $|\mathbf{q}| \to 0$ [46, 47]. A more direct approach to discriminate between insulating and conducting states, e.g., computing the Drude weight, is not possible since it requires the knowledge of excited states.

Analogously, spin order in the system is associated to a peak in the spin-spin correlations defined as:

$$S(\mathbf{q}) = \frac{1}{L}\sum_{R,R'}\langle S^z_R S^z_{R'}\rangle e^{i\mathbf{q}\cdot(\mathbf{R}-\mathbf{R}')}, \tag{11}$$

where $S^z_R$ is the spin operator along the $z$ direction, i.e., $S^z_R = 1/2(c^{\dagger}_{R,\uparrow}c_{R,\uparrow} - c^{\dagger}_{R,\downarrow}c_{R,\downarrow})$.

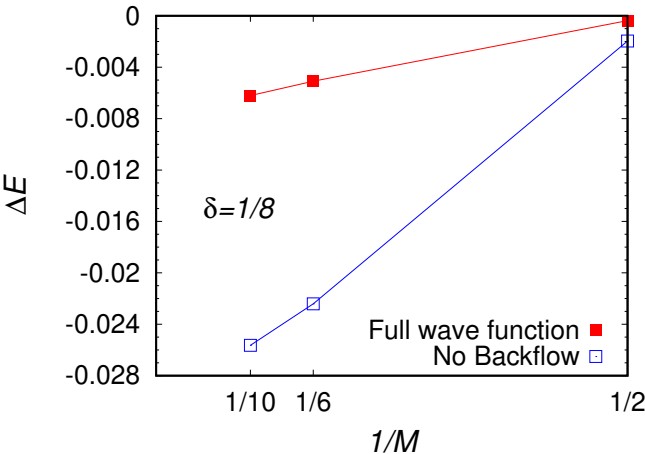

Figure 2: Energy gain (per site) $\Delta E = E_{\text{stripe}} - E_{\text{homogen}}$ of the striped state of wavelength $\lambda = 8$, with respect to the homogeneous superconducting one, as a function of the number of legs $M$, for lattices of size $L = M \times (8 \times M)$. Data are reported for $U/t = 8$ and $\delta = 1/8$, for the cases with and without backflow correlations in the variational state.

Finally, the presence of superconductivity can be assessed by looking at the pair-pair correlations, i.e., a correlation function between singlets on rungs at distance $x$, defined as

$$D(x) = \langle \Delta(x+1)\Delta^{\dagger}(1) \rangle, \tag{12}$$

where

$$\Delta^{\dagger}(x) = c^{\dagger}_{x,1,\uparrow} c^{\dagger}_{x,2,\downarrow} - c^{\dagger}_{x,1,\downarrow} c^{\dagger}_{x,2,\uparrow} \tag{13}$$

is a vertical singlet located on the rung between sites of coordinates $(x, 1)$ and $(x, 2)$. Here, we explicitly denoted the coordinates of the site $R$, i.e., $c^{\dagger}_{R,\sigma} \equiv c^{\dagger}_{x,y,\sigma}$.

Our simulations are performed with periodic boundary conditions along the $x$ direction, while, in the $y$ direction, they are taken to be open for $M = 2$ and periodic for $M = 6$, for $M = 10$, and for 2D lattices. For $M = 2$, open boundary conditions along the rungs are necessary, in order to avoid a double counting of the intra-rung bonds.

## 3 Results

Let us focus on the possibility that a bond-centered stripe may be stabilized at dopings $\delta = 1/p$, with $p$ even. These cases can, in principle, accommodate a filled stripe of wavelength $\lambda = p$ (the full periodicity of the stripe is $2p$ due to spin), as recently suggested in Ref. [31], where the case with $\delta = 1/8$ has been considered in detail. In order to fit the stripe with our cluster with $L = M \times L_x$ sites, we take $L_x$ to be multiple of $2p$.

First of all, in Fig. 2, we plot the energy gain $\Delta E = E_{\text{stripe}} - E_{\text{homogen}}$, where the energies are intended per site, between a homogeneous superconducting state and a striped one for the 2-, 6-, and 10-leg ladders. In order to properly scale to the two-dimensional case, we preserve the aspect ratio of the ladder considering $L = M \times (8 \times M)$. We observe that the energy gain of the striped state with respect to the uniform one increases linearly with the system size. We mention that the less accurate variational wave function without backflow correlations has a larger energy gain with respect to the case in which these correlations are included. Still, even in the most accurate calculations, there is a finite (and relatively large) energy gain

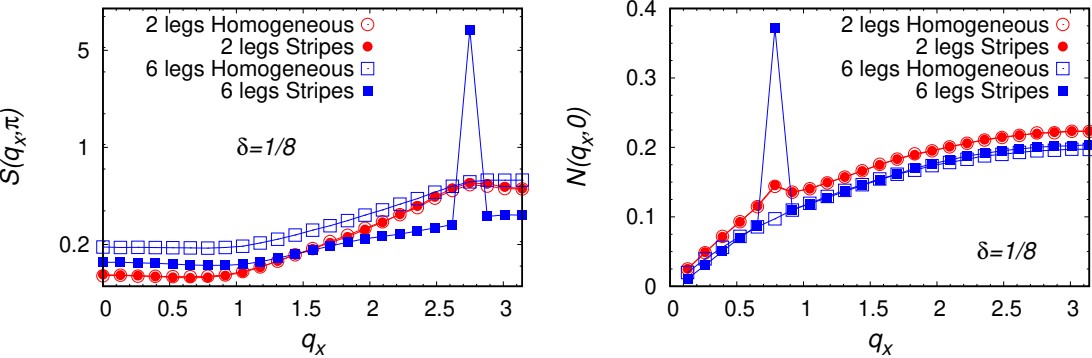

Figure 3: Left panel: Spin-spin correlations $S(q_x, \pi)$ on a semi-log scale, as a function of $q_x$ at doping $\delta = 1/8$. Results are reported for the homogeneous wave function (empty symbols) and for the striped one with wavelength $\lambda = 8$ (full symbols), on a 2-leg ladder with $L_x = 48$ (red circles) and on a 6-leg ladder with $L_x = 48$ (blue squares). Data are shown for $U/t = 8$. Right panel: Same as in the left panel, but on a linear scale for the static structure factor $N(q_x, 0)$.

when charge and spin modulations are considered. Even though a reliable extrapolation to the thermodynamic limit is not easy to perform, we verified that a striped state can be stabilized also in a square-lattice geometry, as we checked for a $L = 16 \times 16$ lattice, observing an energy gain of about $5 \times 10^{-3}$.

In Fig. 3, we report the density-density correlations $N(\mathbf{q})$ and the spin-spin correlations $S(\mathbf{q})$ at doping $\delta = 1/8$, for both the 2-leg and the 6-leg case, comparing the results for a homogeneous wave function and for a striped state. For the 2-leg case, the correlation functions are the same for both wave functions, in agreement with the energy results of Fig. 2, where it is shown that almost no energy gain comes from the explicit inclusion of stripes in the variational state. On the contrary, for the 6-leg case, the striped wave function is characterized by a peak in $N(\mathbf{q})$ at $\mathbf{Q} = (\pi/4, 0)$, in agreement with a periodicity $\lambda = 8$ in the $x$ direction and no charge modulation along the $y$ direction. A peak appears also in the spin-spin correlations $S(\mathbf{q})$ at a vector $\mathbf{Q} = (7\pi/8, \pi)$. Indeed, the spin modulation described in Eq. (7) is antiferromagnetic along the $y$ direction, while it is antiferromagnetic, with an additional modulation of periodicity $2\lambda = 16$, along the $x$ direction. We remark that the peak in $S(\mathbf{q})$ is much stronger than the one in $N(\mathbf{q})$, suggesting that spin modulation plays a major role with respect to charge modulation. We would like to emphasize that, by setting $\Delta_c = 0$ in Eq. (6) and optimizing only $\Delta_{AF}$, we recover the striped state, with peaks in both $S(\mathbf{q})$ and $N(\mathbf{q})$, while, setting $\Delta_{AF} = 0$ in Eq. (7) and optimizing only $\Delta_c$, the stripe is not stable and the optimized state converges to the homogeneous one. This fact implies that spin modulation plays a major role in stripe formation.

As suggested in Ref. [31], our results show that the filled striped state with periodicity $\lambda = 8$ in the charge sector is the appropriate one for the Hubbard model at doping $\delta = 1/8$. The half-filled striped state with periodicity $\lambda = 4$, which has been previously suggested for the $t-J$ model [10, 12], is not stable upon optimization in the Hubbard model. This observation remains valid also for a larger value of the Coulomb repulsion, i.e., $U/t = 16$. In addition to charge and spin modulation, our variational state can also include pairing, either homogeneous or with the modulation of Eq. (8). Both states can be stabilized in the variational state even if neither of them leads to an energy gain with respect to the striped state with no pairing that is shown in Fig. 2.

We have then compared homogeneous and striped wave functions at dopings $\delta = 1/p$,

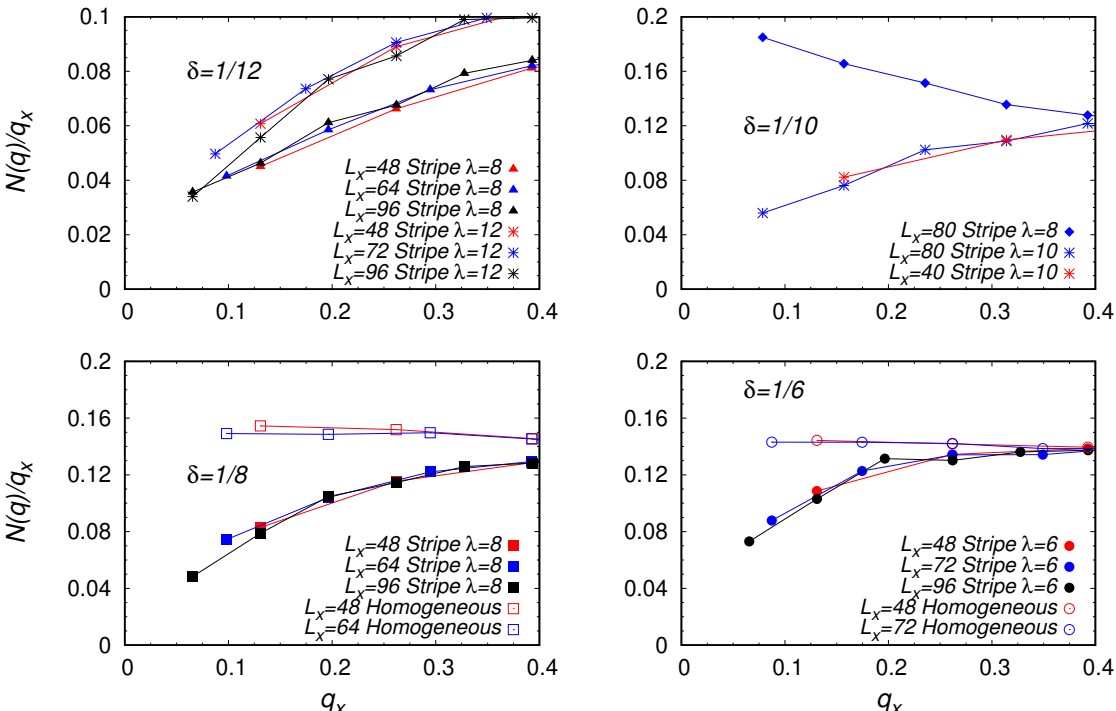

Figure 4: $N(\mathbf{q})/q_x$, as a function of $q_x$ at $q_y = 0$, on lattice sizes $L = 6 \times L_x$. Results are reported at $\delta = 1/12$ (top left) for the optimal striped state with $\lambda = 8$ (up triangles) and for the striped state with $\lambda = 12$ (stars); at $\delta = 1/10$ (top right) for the optimal striped state with $\lambda = 8$ (diamonds) and for the striped state with $\lambda = 10$ (stars); at $\delta = 1/8$ (bottom left) for the homogeneous wave function (empty squares) and for the optimal striped state with $\lambda = 8$ (full squares); at $\delta = 1/6$ (bottom right) for the homogeneous wave function (empty circles) and for the optimal striped state with $\lambda = 6$ (full circles).

with $p = 12$, 10, and 6, that are also relevant for the phase diagram of cuprate superconductors. We observe that at dopings $\delta = 1/12$ and $\delta = 1/10$, the best striped state has wavelength $\lambda = 8$, like in the $\delta = 1/8$ case. Striped states of wavelength $\lambda = 12$ and $\lambda = 10$ can be stabilized as local minima, with an energy difference with respect to the best stripe of the order of $10^{-3}$. On the contrary, the striped state of wavelength $\lambda = 8$ cannot be stabilized at doping $\delta = 1/6$, where, instead, a stripe of wavelength $\lambda = 6$ characterizes the ground state. In Table 1, we present a summary of the energies per site of homogeneous and striped states. Even if our accuracy in the ground-state energies is not as good as what has been reported in Ref. [31], we are expected to capture the correct ground-state behavior, since we simultaneously optimize a large number of variational parameters, that can capture the correct ground-state behavior among a wide range of possible quantum phases. Our results show that the energy gain due to striped phases with respect to homogeneous phases is becoming larger when approaching half filling. Furthermore, we observe that at all densities we can stabilize a finite BCS pairing, both uniform and modulated, within the striped state, even if this pairing is reduced with respect to the uniform case.

The insulating or metallic nature of the ground state can be seen in the small-$q$ behavior of the static structure factor $N(\mathbf{q})$. In Fig. 4, we show $N(\mathbf{q})/q_x$ for homogeneous states (empty symbols) and for the striped ground states (full symbols) at doping $1/p$, with $p = 12$, 10, 8, and 6. We observe that in the homogeneous cases, as well as for the optimal striped state at $p = 10$

Table 1: Energies per site for the homogeneous state, the best striped state, as well the energy gain $\Delta E = E_{\text{stripe}} - E_{\text{homogen}}$ between the best striped state and the homogeneous one, at dopings $1/p$ with $p = 6, 8, 10,$ and 12. Error-bars are estimated in order to take into account the weak size dependence of the energies; calculations have been performed on the following lattices: $L = 6 \times L_x$ with $L_x = 48, 72,$ and 96 at $p = 6$; $L = 6 \times L_x$ with $L_x = 48, 64,$ and 96 at $p = 8, 12$; $L = 6 \times 80$ at $p = 10$.

| $p$ | $E/t$ (Homogeneous) | $E/t$ (Best stripe) | $\Delta E$ |
|---|---|---|---|
| 12 | -0.6661(1) | -0.6726(1) ($\lambda = 8$) | -0.0065(1) |
| 10 | -0.6966(1) | -0.7027(1) ($\lambda = 8$) | -0.0061(1) |
| 8 | -0.7436(1) | -0.7483(1) ($\lambda = 8$) | -0.0047(1) |
| 6 | -0.8191(1) | -0.8207(1) ($\lambda = 6$) | -0.0016(1) |

(with $\lambda = 8$), $N(\mathbf{q}) \simeq q_x$ at small $q_x$, clearly indicating that the state is metallic. The latter result can be explained by the fact that the optimal stripe is not commensurate with the doping, since an even number of holes cannot be accommodated in each of the $L_x/16$ periods in the spin modulation. The results for the striped ground states at $p = 8$ and 6 are instead compatible with an insulating ground state, with the behavior of $N(\mathbf{q})/q_x$ extrapolating to zero when the lattice size increases. These striped states accommodate $2M$ holes in each of the $L_x/2p$ periods in the spin modulation. At $p = 12$, the situation is less clear, but more compatible with a metallic behavior for the optimal striped state with $\lambda = 8$ and with an insulating behavior for the filled stripe with $\lambda = 12$. In this case, even if the optimal striped state is not filled, it is still commensurate with the doping since, on a 6-leg ladder, an even number of holes can be accommodated in each of the $L_x/16$ periods in the spin modulation (with an unequal distribution of holes in different legs). The ordered nature of these striped states can be clearly seen in the emergence of peaks in the charge and in the spin structure factors. Given the wavelength $\lambda$ of the optimal stripe, $N(\mathbf{q})$ has a peak at $\mathbf{Q} = (2\pi/\lambda, 0)$ and $S(\mathbf{q})$ has a peak at $\mathbf{Q} = [\pi(1 - 1/\lambda), \pi]$, as reported in Fig. 5 for $p = 12, 8,$ and 6. The divergence of these peaks with the system size is also reported in Fig. 6, with both $\lim_{L \to \infty} N(\mathbf{Q})/L$ and $\lim_{L \to \infty} S(\mathbf{Q})/L$ decreasing upon increasing doping.

We mention that the stripe wavelength increases when approaching half filling, as we have verified at doping $\delta = 1/16$, where the optimal striped state is metallic with $\lambda = 12$. Here, different stripe wavelengths with a long periodicity are close in energy, as expected when the system is prone to phase separation. Indeed, at the accuracy level of variational Monte Carlo, a region of phase separation is predicted close to half filling [38, 48, 49].

Generic dopings are more difficult to treat, because they cannot be exactly reproduced on different lattice sizes; however, some predictions can be formulated also in this case. In particular, for the density range between dopings $\delta = 1/12$ and $\delta = 1/8$, where we can assume that a possible striped state has to have a wavelength $\lambda = 8$, since this is the optimal one for the two extremal cases. Given this assumption, we can consider a generic incommensurate doping: for example, doping $\delta = 0.104$ that can be approximately obtained on both $L = 6 \times 48$ and $L \times 64$ lattice sizes. Here, the ground state is metallic in analogy with the $p = 10$ case, with an energy gain of the order of $6 \times 10^{-3}$ when stripes are included. A similar analysis can be performed also for a generic doping, in the range between $\delta = 1/8$ and $\delta = 1/6$, leading to analogous conclusions on the metallic nature of the ground state. In this case, we cannot exclude that, for a sufficiently large lattice size, a suitable striped state leading to an insulating ground state may be found, since the optimal stripe wavelength shifts from $\lambda = 8$ to $\lambda = 6$ at $\delta \simeq 0.13$. Finally, we observe that for dopings $\delta \gtrsim 0.20$, no stripe order can be stabilized in the

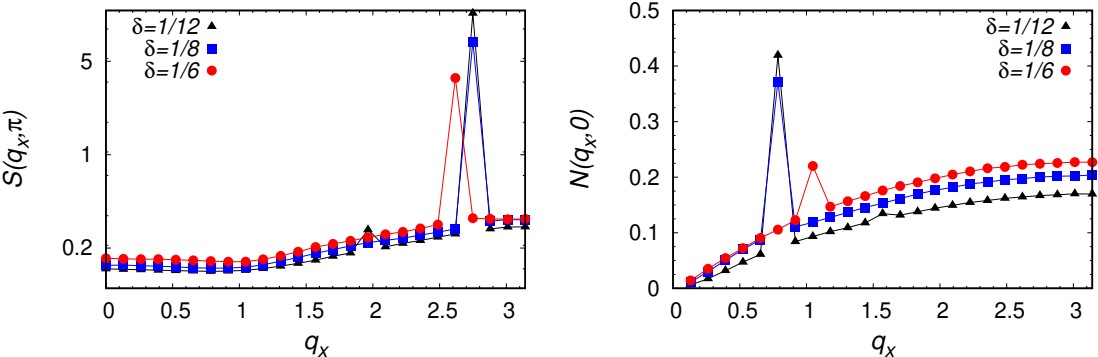

Figure 5: Left panel: Spin-spin correlations $S(q_x, \pi)$ on a semi-log scale, as a function of $q_x$ at dopings $\delta = 1/12$ (black triangles), $\delta = 1/8$ (blue squares), and $\delta = 1/6$ (red circles), for the best striped state, that has wavelength $\lambda = 8$ at $\delta = 1/12$ and at $\delta = 1/8$ and has wavelength $\lambda = 6$ at $\delta = 1/6$. Data are reported on a 6-leg ladder with $L_x = 48$ at $U/t = 8$. Right panel: Same as in the left panel, but on a linear scale for the static structure factor $N(q_x, 0)$.

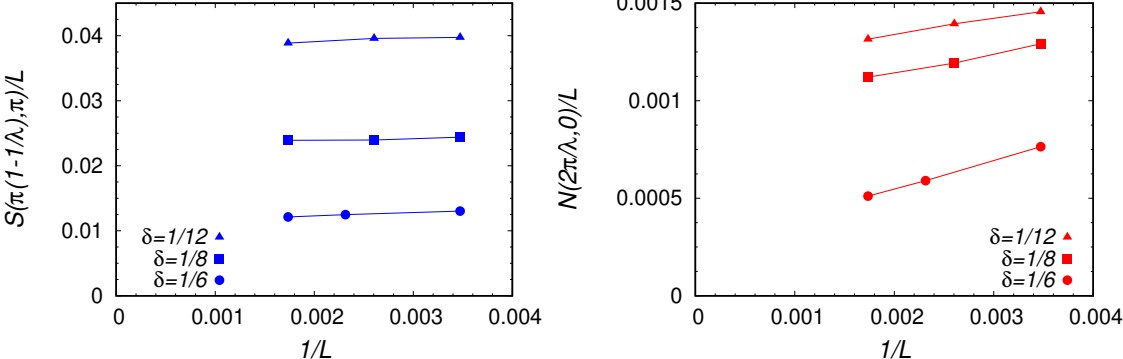

Figure 6: Left panel: Spin-spin correlations $S(\mathbf{q})$ at the pitch vector $\mathbf{Q} = [\pi(1 - 1/\lambda), \pi]$, divided by the lattice size $L$, as a function of $1/L$ at doping $\delta = 1/p$, with $p = 12$ (triangles), 8 (squares), 6 (circles). Right panel: Density-density correlations $N(\mathbf{q})$ at the pitch vector $\mathbf{Q} = (2\pi/\lambda, 0)$, divided by the lattice size $L$, as a function of $1/L$ at doping $\delta = 1/p$, with $p = 12$ (triangles), 8 (squares), 6 (circles). The value of $\lambda$ for each doping is indicated in Table 1. Results are shown for 6-leg ladders and $U/t = 8$.

wave function, with the ground state being a homogeneous superconductor up to $\delta_c \simeq 0.27$, as reported in Ref. [18].

Finally, we show in Fig. 7 the superconducting pair-pair correlations $D(x)$, defined in Eq. (12). We remark that the presence of a finite BCS pairing in the variational state does not necessarily lead to superconductivity, which presence or absence is seen in the long-range behavior of the pair-pair correlations $D(x)$. For instance, the Mott insulating state at half filling is characterized by a finite BCS pairing in the variational state and vanishing pair-pair correlations at large distances [26]. From the analysis of the results, we deduce that superconductivity, in the presence of stripes, disappears for all the three dopings $\delta = 1/p$ with $p = 12$, 8, and 6, with $D(x) \simeq 0$ for large enough $x$. This result is in agreement with the insulating (or weakly metallic) nature of the striped states. Once we move away from these dopings, the system is metallic and superconducting correlations are present, even if suppressed with

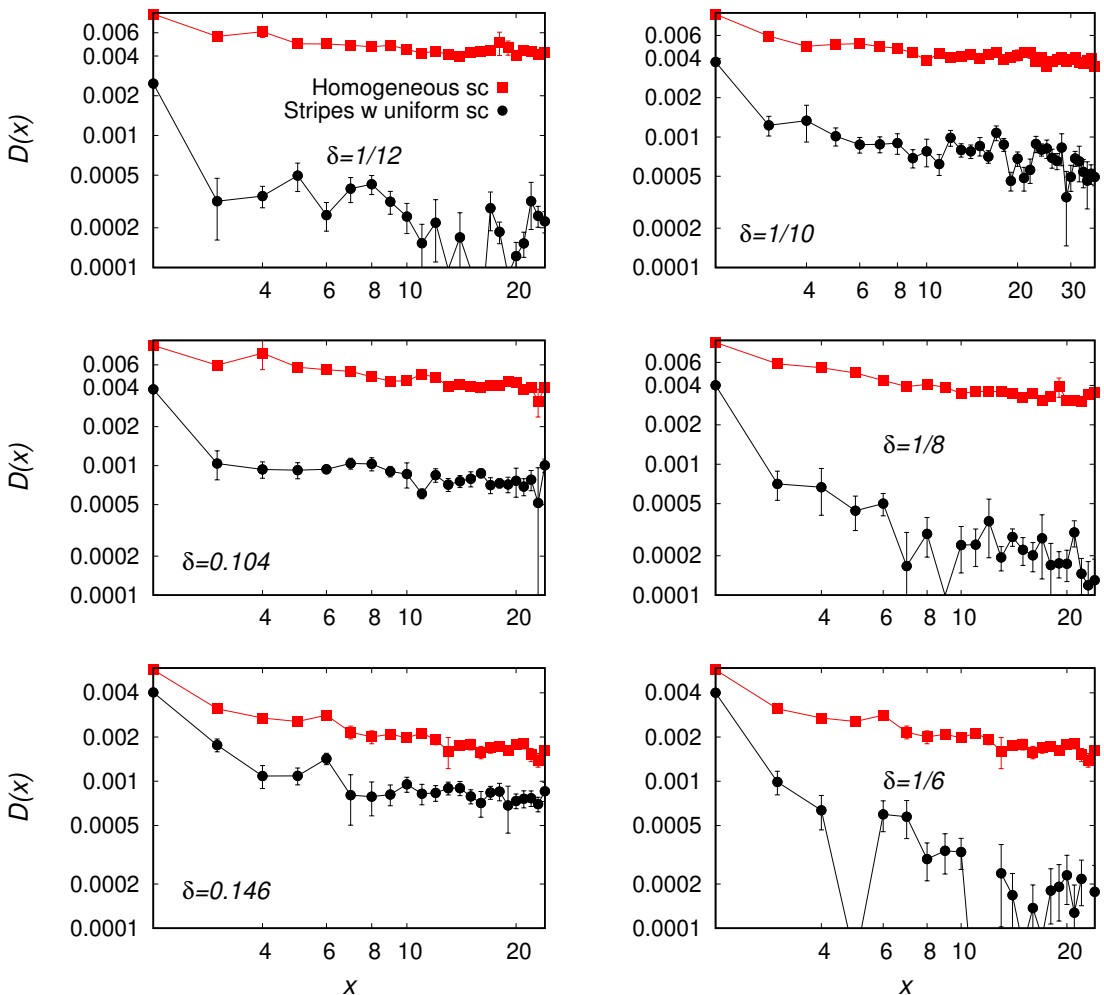

Figure 7: Superconducting pair-pair correlations $D(x)$ as a function of $x$ on the 6-leg ladder at increasing dopings $\delta = 1/12$, $1/10$, $0.104$, $1/8$, $0.146$, and $1/6$, from top left to right bottom. Two different wave functions are employed: the homogeneous one of Eqs. (3) and (4) (red squares), and the striped one of Eq. (5) with uniform pairing (black circles). The stripe wavelength is $\lambda = 8$ at $\delta = 1/12$, $1/10$, $0.104$, and $1/8$, while it is $\lambda = 6$ at $\delta = 0.146$ and $1/6$. Data are reported at $U/t = 8$ and $L = 6 \times 48$ (except at doping $\delta = 1/10$, where they are reported on a $L = 6 \times 80$ lattice size), on a log-log scale in order to highlight the power-law decay.

respect to the homogeneous case. In particular, in Fig. 7, we present results for $p = 10$ and for two generic dopings, $\delta = 0.104$ and $\delta = 0.146$. Our results in the presence of stripes are shown with uniform superconductivity, but they would be very similar also for the modulated one of Eq. (8).

## 4 Conclusions

In this work, we studied the possibility to stabilize charge and spin modulations (stripes) in the Hubbard model in a non vanishing doping range, and their coexistence with superconductivity. To this end, we employed variational wave functions that contain both Jastrow and backflow correlations. Even though the calculations have been performed on ladder geometries with

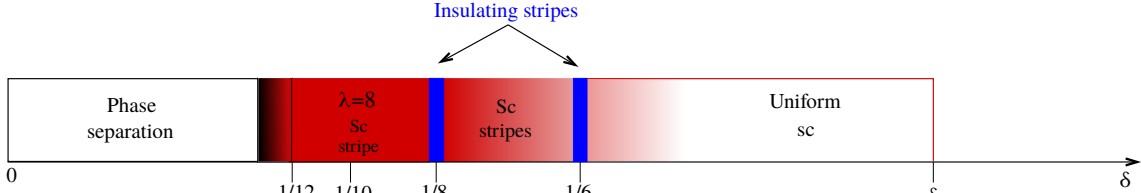

Figure 8: Schematic phase diagram of the doped repulsive Hubbard model at $U/t = 8$ on a 6-leg ladder. The graduated shading indicates the progressive weakening of the stripe order.

$L = M \times L_x$ sites (with $M = 2$, 6, and 10, and $L_x \gg M$), the results are expected to be relevant also for the thermodynamic limit. Indeed, as discussed in Ref. [31], the DMRG calculations performed on ladders with $M = 4$ and 6 already contain the qualitatively correct features of the two-dimensional limit. In addition, our results on the $16 \times 16$ cluster (not shown) are compatibles with the ones obtained on ladders. We also mention that we have performed some preliminary calculations, based on Green's Function Monte Carlo [50] with the Fixed-Node approximation [51], confirming that at dopings $\delta = 1/p$, with $p$ even, the best variational state contains stripes and confirming that at doping $\delta = 1/12$ the best striped state has wavelength $\lambda = 8$.

Our main results are summarized in Fig. 8. At doping $\delta = 1/8$ and $1/6$, we obtain insulating stripes of wavelength $\lambda = 1/\delta$ (the full periodicity is doubled due to spin). The stripe with $\lambda = 8$ appears to be particularly stable, since it corresponds to the lowest-energy state also for $\delta = 1/10$ and $\delta = 1/12$. In the former case, the ground state shows a metallic behavior with finite superconducting pair-pair correlations, which is compatible with the mismatch between the doping level and the periodicity of the stripe; furthermore, we have verified that also for the 10-leg case the best striped state at $\delta = 1/10$ is metallic with a wavelength of $\lambda = 8$. In the latter case, instead, it is not so clear whether the ground state is metallic or insulating, since $N(\mathbf{q})/q_x$ may extrapolate either to zero (insulating) or to a small finite value (metallic); moreover, superconducting correlations decay rapidly to zero with the distance $x$. Intermediate (incommensurate) dopings are much more difficult to assess; however, we obtain that a metallic stripe with $\lambda = 8$, coexisting with superconductivity, can be stabilized in the whole doping range between $\delta = 1/12$ and $\delta = 1/8$. Also for $\delta > 1/8$ we have indications of superconducting stripes, even if we cannot exclude that, for a sufficiently large lattice size, a suitable striped state leading to an insulating ground state may be found. Finally, for $0.20 \lesssim \delta \lesssim \delta_c = 0.27$, uniform superconductivity is present. Our results indicate that the best place to observe a suppression of superconductivity due to stripes is indeed $1/8$ doping. In addition, we foresee that sizable effects should be also visible at $\delta = 1/6$.

One important aspect of our calculations is that stripe order is driven by spin and not by charge. Indeed, as discussed in the text, if we only include modulations in charge, but not in spin, no stripe order can be stabilized in the variational optimization; instead, a striped ground state is reached when only spin modulations are included in the variational state.

# Acknowledgements

We thank S. Sorella for useful discussions.

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
