# Peer review of "Metallic and insulating stripes and their relation with superconductivity in the doped Hubbard model"

_SciPost Physics, doi:SciPost Phys. 7, 021 (2019)_

## Round 1 · Referee Report · Anonymous (Referee 1) · 2019-6-12

Report

Report

In this paper the authors study the doped Hubbard model by means of the variational Monte Carlo (VMC) approach based on Jastrow-Slater wave functions, improved by the inclusion of backflow correlations. They find an insulating stripe ordered state with a period 8 in the charge order at 1/8 doping, in agreement with other numerical methods from other recent works. The period 8 stripe is also the lowest energy state at smaller doping (1/10 and 1/12), whereas at doping 1/6 they find an insulating period 6 stripe. In between these dopings the stripes exhibit strong superconducting correlations. The authors report that the spin modulation is crucial in order to obtain the stripes in this work; whereas without the spin modulation uniform states are favored (which is the reason why uniform states have been predicted in previous works based on VMC). They also find that stripes with in-phase pairing are lower in energy than the ones with anti-phase pairing, and that uniform states become favorable for dopings larger than 0.2.

Understanding the ground state phase diagram of the 2D Hubbard model has been a major and central challenge since many decades. While 10 years ago there was a huge discrepancy between the results obtained with various numerical approaches, the situation has changed in recent years and there is now a growing consensus, at least regarding certain aspects of the phase diagram. With this work the authors provide an important contribution in this context, in particular they strengthen the recent result that the ground state of the doped 2D Hubbard model in the strongly correlated regime is a stripe state, and not a uniform d-wave superconducting state, which has been an open and very controversial question for many years. Having additional support from VMC regarding this question is another milestone towards a full solution of the 2D Hubbard model.

Besides this main result, the authors provide other interesting results, e.g. that the period 8 stripe is also the lowest energy state at smaller dopings, and the nature of the stripes (insulating/metallic/superconducting) for different dopings. It is also very interesting to finally understand, why stripes have not been predicted in previous VMC studies (because in previous works the spin order has not been included in the variational optimization). The paper is also well written and the results are presented in a clear way.

For these reasons I can definitely recommend publication of this interesting article in SciPost. I only have minor points for improvement and questions attached below.

Questions and comments:

1. The arrangement of the figures could be improved; there are sometimes several pages between a figure and the text describing it. A possible solution could be to make the figures more compact, e.g. putting the two panels in Figs. 3, 5, 6 next to each other rather than on top of each other, and then combine more figures on one page.

2. The caption of Fig. 4 is a bit cumbersome to read with the listing of all plot symbols. Putting a legend in each figure panel would make it more readable and more compact.

3. I did not quite understand the following statement in the conclusions part: "A weakly metallic behavior with no superconducting correlations may be related to the proximity to the phase separation region that, within the variational Monte Carlo accuracy, is present close to half filling."
Could the authors explain in more detail what they mean by this statement?

4. Is it in principle possible to further improve the results and check the stability of the stripe order using additional Lanczos steps and combination with fixed-node Monte Carlo? (I am not requesting this for this work, but just to know if there would be further room for improvement).

  • validity: -
  • significance: -
  • originality: -
  • clarity: -
  • formatting: -
  • grammar: -

Author:  Luca Fausto Tocchio  on 2019-07-16  [id 561]

(in reply to Report 1 on 2019-06-12)
Category:
answer to question

We thank the Referee for his/her careful report and for recommending our paper for publication. Here, we provide an answer to his/her questions and comments:

1) We agree with the Referee that putting two panels next to each other is a much better option. We prepared a new version of figures 3, 5, and 6 for the revised manuscript.

2) We agree with the Referee and follow his/her suggestion, putting a legend in each figure panel.

3) Indeed, the statement is just a speculation to reconcile the (weakly) metallic nature of the system at doping 1/12, with the fast decay of the superconducting correlations, as instead observed where the system is insulating. Since this statement is not relevant for the conclusions, we simply decided to drop the sentence.

4) The Referee is right. The stability of the stripe ordered state can be further checked by improving the variational state. As a preliminary check, we verified that at doping 1/12 the best striped state is the one with wavelength $\lambda=8$, also after applying the fixed-node Monte Carlo method. We have also verified that, at dopings 1/12, 1/8, and 1/6, the best variational state is a striped one and not an uniform one, even after applying the fixed-node Monte Carlo method.

We report here the energies, computed within the fixed-node approximation (FN) on a $L=288$ lattice size, when the starting variational state is homogeneous and when the starting state is the best striped one, as well as their difference $\Delta E$ (to be compared with data in Table I):

doping $E/t$ (FN homogeneous) $E/t$ (FN best stripe) $\Delta E$ 1/12 -0.67309(3) -0.67820(3) 0.00511(6) 1/8 -0.74998(2) -0.75398(2) 0.00400(4) 1/6 -0.82338(2) -0.82576(2) 0.00238(4)

Since these are very preliminary calculations, which do not change the conclusions of the work, we prefer not to include them in the manuscript. Nevertheless, we added a sentence on that in the concluding part.

---

## Round 1 · Referee Report · Anonymous (Referee 2) · 2019-6-14

Report

In this manuscript, the authors describe variational Monte Carlo
calculations on the two-dimensional Hubbard model treated in ladder
geometry (actually, toroidal geometry with the x- spatial direction
taken to be much larger than the transverse y-direction). The authors
confirm and build on previous numerical results (in particular, those
in Ref. [29]) with an independent method. These calculations find
that completely filled and thus insulating stripe configurations
dominate at moderate to strong Coulomb interaction strength and 1/8
doping away 1/2 filling. In addiition the authors' more extensive
treatment of the effect of changing the band filling on the stripe
configurations and pairing strength adds to the picture and makes
important progress towards determining the ground-state phase diagram
as a function of doping for the two-dimensional Hubbard model, as
depicted by the authors in Fig. 8. In particular, that filled stripes
and thus non-superconducting behavior is dominant at particular
commensurate band fillings, but that strong superconducting
correlations can occur at more generic fillings, and that the stripe
order melts at sufficiently large doping is an important result.

Aside from a few details mentioned below, I feel that the work in the
has been very carefully carried out and that the results are very
clearly depicted and described. Thus, I strongly recommend that this
work be accepted for publication on SciPost Physics.

I do, however, also have the following detailed comments, most of
which are relatively minor and which I hope would improve the
readability of the manuscript and the support for the calculation
when/if implemented:

1) The authors have treated lattices (primarily) with L_y = 2, 6,
10. Is there a particular reason that L_y = 4, 8 were left out? In
particular, L_y = 4 might be useful to compare directly with other
work such as that in Ref. [29].

2) I'm not convinced that the low-q behavior of the charge structure
factor is such a definitive probe of the metallicity; it
esssentially just probes the behavior of the integral of the
charge-charge correlation function. It cannot distinguish between
normal metallicity and superconductivity. A more transport-oriented
measure such as the Drude weight or the electric susceptibility
would be better. In addition, for the low-q behavior of the
structure factor, it would be good to carry out a finite-size
scaling by studying the behavior of the lowest non-zero q point as
a function of L_x.

3) I think that some details of the authors description of the
variational energies, Fig. 2 and Table 1, can be tightened up. It
seems to me that the energies quoted must be intensive, i.e.,
energy per site. If so, this should be stated explicitly.
In Table 1, how are the errors in the energies estimated? (It is a
little suspect that they are all exactly the same on an absolute
scale, for all states and fillings.
Also, can the variational energies be compared to other variational
calculations, such as the DMRG calculations in Ref. [29]? (Here
considerations of the effect of boundary conditions as well applied
extrapolations would presumably complicate the matter.)

4) In the variational state, it is not completely clear to me what
effect allowing/turning on the pairing correlation have on the
results. On the one hand, we have statements that the metallicity
and pairing are necessarily suppressed in the filled stripe
states. On the other, we have statements (e.g., on p. 10, last
paragraph) that various pairing correlations can be induced
without significantly changing the energy. Presumably the latter
statement is only a description of the initial variational state,
which is then modified by the Jastrow factors, Monte Carlo
optimization, and back flow. But a more clear statement could be
made about the instrinsic robustness of the pairing in the
variational state.

5) I feel that the analysis of the pairing correlations, Fig. 7, is
a little too casual. In particular, what is important is the
exponent of the power-law decay at moderate to large distance,
assuming that they can be sufficiently accurately calculated at
these distances. For the stripe states, the pairing is always
suppressed, even at short distance, but this can be viewed as a
prefactor that multiplies the asymptotics. The authors should
comment more directly and quantitatively on the asymptotics, if
possible, or describe why the asymptotics cannot be reliably
extracted if not.

6) In Fig. 2, it would be better to label the x axis with 1/10, 1/6,
1/2, rather than decimal notation and/or list the treated M-values
explicitly.

7) It might be useful to remind the reader of the treated optimal
stripe wavelengths in the legend or caption of Figs. 5, 6, 7, and
Table 1 to make interpretation easier for the reader, even though I
realize that the optimal stripe wavelengths are mentioned multiple
times in the text.

  • validity: high
  • significance: high
  • originality: good
  • clarity: high
  • formatting: excellent
  • grammar: excellent

Author:  Luca Fausto Tocchio  on 2019-07-16  [id 562]

(in reply to Report 2 on 2019-06-14)
Category:
answer to question

We thank the Referee for his/her careful report and for recommending our paper for publication. Here, we provide an answer to his/her questions and comments:

1) Our choice of selecting $L_y$=6 as the geometry in which most of our calculations are performed is motivated by comparisons with previous DMRG results that have been obtained on this geometry. In addition, in Ref. [31] (that was Ref. [29] in the previous version), the comparison between DMRG and other numerical approaches suggested that $L_y$=6 is already large enough to capture many aspects of the 2D limit.

Within our variational approach, the case with no stripes and $d$-wave superconductivity has Dirac points at $(\pm \pi/2, \pm \pi/2)$; then, in order to have a well-defined uncorrelated state $|\Phi_0\rangle$, periodic-boundary conditions along $y$ imply that $L_y=4n+2$ ($n$ being an integer). Otherwise, $L_y=4n$ would imply anti-periodic-boundary conditions, which is not commonly used in other approaches. A more detailed comment on this point can be found in the Variational Monte Carlo section of Ref. [18] (in the actual version of the manuscript).

2) The Referee is correct when saying that the low-$q$ behavior of the charge structure factor cannot distinguish between a normal metal and a superconductor. In fact, we use this probe only to discriminate between insulating and conducting states and we compute superconducting correlations to check whether the system is a superconductor.

Unfortunately, computing the Drude weight within the variational approach is impossible, since it would require the knowledge of excited states. We added a sentence on that in the revised version of the manuscript.

Concerning the second part of the point raised by the Referee, we disagree with him/her on the point that no finite-size scaling of the low-$q$ behavior is performed. The results presented in Fig. 4 are discussed for different values of $L_x$, whenever different values of $L_x$ commensurate with the stripe length are available. Unfortunately, a proper fit of the lowest non-zero $q$ point as a function of $L_x$ cannot be performed when only three sizes are available and we agree that, in some regimes, it may be hard to discriminate between an insulating an a conducting behavior from the data shown in Fig. 4. In order to improve the presentation of the results, we have prepared an additional figure (fig_reply.pdf), attached to this reply, where we show the lowest non-zero $q$ point as a function of $1/L_x$ for four selected cases. Our data are more compatible with an insulating behavior for the optimal stripe of wavelength $\lambda=8$ at doping 1/8 and for the stripe of wavelength $\lambda=12$ at doping 1/12, while they are more compatible with a (weakly) conducting behavior for the optimal stripe of wavelength $\lambda=8$ at doping 1/12. Results at doping 1/6 are less clear, but since the stripe wavelength is $\lambda=6$ we expect the system to be insulating. We do not further consider the homogeneous cases and the optimal stripe of wavelength $\lambda=8$ at doping 1/10, since these cases are clearly conducting.

3) The energies are intensive, we will mark this point more clearly in the manuscript.

Each entry of Table I has an energy that slightly depends on the lattice size. However, this dependence on the lattice size is pretty weak and an error bar of the order of $10^{-4}$ takes into account the variations that are due to the lattice size. In this respect, the error bar is inserted "by hand" in order to take into account the lattice size dependence. Given the lattice size, the statistical errors are instead estimated with the usual techniques and are of the order of $10^{-5}$. We will clarify this point in the revised version of the manuscript.

At doping 1/8 we obtain a best estimate of the ground state energy of about $E/t=-0.748$, while DMRG obtains an estimate of about $E/t=-0.763$, following Ref. [31] (that was Ref. [29] in the previous version). As noticed by the first Referee, our energies may be improved by using the fixed-node projection technique, as reported below. Still, the aim of this paper is not to get energies which are competitive with the DMRG ones, but to capture the correct physical behavior. In this regard, we think that, even having (slightly) higher energies with respect to DMRG, the ground-state properties are correctly described, since our variational states include a large number of variational parameters, that are simultaneously optimized to capture the ground-state behavior, among a wide range of possible quantum phases. We added a sentence in the "Results" section on that.

We report here the energies, computed within the fixed-node approximation (FN) on a $L=288$ lattice size, when the starting variational state is homogeneous and when the starting state is the best striped one, as well as their difference $\Delta E$ (to be compared with data in Table I):

doping $E/t$ (FN homogeneous) $E/t$ (FN best stripe) $\Delta E$ 1/12 -0.67309(3) -0.67820(3) 0.00511(6) 1/8 -0.74998(2) -0.75398(2) 0.00400(4) 1/6 -0.82338(2) -0.82576(2) 0.00238(4)

4) We think that the previous version of the manuscript was not completely clear on this aspect. The presence of a finite pairing $\Delta_x$ and $\Delta_y$ in the BCS Hamiltonian (that defines the uncorrelated wave function) does not directly imply that there are finite pair-pair correlations, as defined in Eq.(12), in the correlated state. For example, within the Mott insulator at half filling, a finite $\Delta$ gives an energy gain, but superconducting correlations are suppressed because of the Jastrow factor, in agreement with the proposed Resonant Valence Bond state. Here, within the striped state, a finite value of the variational parameter $\Delta$ may be found (which leads to a tiny energy gain), but still pair-pair superconducting correlations can be suppressed, as it happens for instance at dopings 1/12, 1/8, and 1/6, see Fig. 7. In the revised version of the paper, we made clear statements on this issue.

5) The most important information of Fig. 7 is the fact that pairing correlations decay rapidly to small values at doping 1/12, 1/8, and 1/6, while they follow the behavior of the uniform case, even if with some suppression, at doping 1/10 and for intermediate incommensurate dopings.

We agree with the Referee that analyzing the asymptotics of the superconducting correlations function may provide further information on the behavior of superconductivity between the uniform and the striped cases. However, it is not easy to obtain a reliable values for the asymptotic behavior (i.e., the values of the exponents) from our data. This analysis would require massive numerical simulations and goes well beyond the scope of the paper, which is to highlight the qualitative difference between cases that have extremely small values of the pairing correlations, like dopings $\delta=$1/12, 1/8, and 1/6, and cases that show only some suppression of superconductivity, with respect to the uniform case, like dopings $\delta=$1/10, 0.104, and 0.146.

6) We changed the $x$-axis label accordingly.

7) We follow the suggestion of the Referee in the revision of the manuscript.

Attachment:

fig_reply.pdf

---

## Round 2 · Referee Report · Anonymous (Referee 1) · 2019-7-17

Report

The authors have revised and improved their paper according to all points of criticisms and I can thus recommend publication of this interesting article in SciPost in its present form.

---

## Round 2 · Referee Report · Anonymous (Referee 2) · 2019-7-30

Report

The authors have adequately addressed all of the points raised in my referee report and in the report of referee 1 and, in addition, have included all of the suggested improvements to the text and figures from both referees.

I recommend that this improved manuscript now be published in this form on SciPost.

---

## Round 2 · Author Response

Dear Editor,

We would like to resubmit the manuscript ``Metallic and insulating stripes and their relation with superconductivity in the doped
Hubbard model'' by L.F. Tocchio, A. Montorsi, and F. Becca to SciPost Physics.

We thank the Referees for their positive reports and for strongly recommending the paper for publication after some clarifications
have been provided. We submitted a reply to both the Referee reports, where we answer the questions raised by them; the revised version of the manuscript includes
all the pertinent modifications.

We also provide a point-to-point list of all the changes.

Sincerely yours,
L.F. Tocchio, A. Montorsi, and F. Becca

---

## Round 2 · List of Changes

List of modifications:

1) We changed Figs. 2, 3, 4, 5, 6, and 7 (and/or their captions) according to the suggestion of Referee 1 (points 1 and 2) and Referee 2 (points 6 and 7); 2) We modified the Conclusions, according to the comments of Referee 1 (points 3 and 4); 3) We changed the "Variational Monte Carlo method" section, where the static structure factor is introduced, following the point 2 raised by Referee 2; 4) We added a few comments in the section "Results" in order to clarify the points 3 and 4 raised by Referee 2. 5) We added two extra references on the t-J model in the "Introduction" section and a related comment on the nature of the pairing term in the variational state in the "Variational Monte Carlo method" section.

---

## Editorial Decision

published